# Is the Hubble Crisis Connected with the Extinction of Dinosaurs?

Leandros Perivolaropoulos 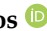

Department of Physics, University of Ioannina, 45110 Ioannina, Greece; leandros@uoi.gr

**Abstract:** It has recently been suggested that a gravitational transition of the effective Newton's constant $G_{eff}$ by about 10%, 50–150 Myrs ago could lead to the resolution of both the Hubble crisis and the growth tension of the standard ΛCDM model. Hints for such an abrupt transition with weaker gravity at times before the transition, have recently been identified in Tully–Fisher galactic mass-velocity data, and also in Cepheid SnIa calibrator data. Here we use Monte-Carlo simulations to show that such a transition could significantly increase (by a factor of 3 or more) the number of long period comets (LPCs) impacting the solar system from the Oort cloud (semi-major axis of orbits $\gtrsim 10^4$ AU). This increase is consistent with observational evidence from the terrestrial and lunar cratering rates, indicating that the impact flux of kilometer sized objects increased by at least a factor of 2 over that last 100 Myrs compared to the long term average. This increase may also be connected with the Chicxulub impactor event that produced the Cretaceous–Tertiary (K-T) extinction of 75% of life on Earth (including dinosaurs) about 66 Myrs ago. We use Monte-Carlo simulations to show that for isotropic Oort cloud comet distribution with initially circular orbits, random velocity perturbations (induced e.g., by passing stars and/or galactic tidal effects), lead to a deformation of the orbits that increases significantly when $G_{eff}$ increases. A 10% increase in $G_{eff}$ leads to an increase in the probability of the comets to enter the loss cone and reach the planetary region (pericenter of less than 10 AU) by a factor that ranges from 5% (for velocity perturbation much smaller than the comet initial velocity) to more than 300% (for total velocity perturbations comparable with the initial comet velocity).

**Keywords:** cosmology; galaxies; Tully–Fisher relation; gravitational transition





## 1. Introduction

The expansion rate of the Universe is predicted by the standard ΛCDM model to have the form

$$H(z) = H_0 \left[ \Omega_{0m}(1+z)^3 + (1 - \Omega_{0m}) \right]^{1/2} \equiv H_0 \, E(z) \qquad (1)$$

where $H_0$ is the Hubble constant and $\Omega_{0m}$ is the matter density parameter. The functional form of $E(z)$ is well constrained by cosmological standard rulers (e.g., the sound horizon scale at recombination probed by Baryon Acoustic Oscillations and the CMB anisotropy power spectrum) and standard candles (e.g., Type Ia Supernovae SnIa) to be consistent with Equation (1) for $\Omega_{0m} = 0.315 \pm 0.007$ [1]. However, when the Hubble constant $H_0$ is measured using local calibrators at low redshifts $z < 0.01$ (e.g., Cepheid stars), the value of $H_0$ is found to be $H_0 = 73.04 \pm 1.04$ km s$^{-1}$ Mpc$^{-1}$ [2,3] higher and at approximately $5\sigma$ tension with the corresponding value $H_0 = 67.4 \pm 0.5$ km s$^{-1}$ Mpc$^{-1}$ [1] obtained using the sound horizon at recombination as a standard ruler calibrated via the the CMB anisotropy spectrum or by Big Bang Nucleosynthesis (BBN) at $z > 1100$. This discrepancy has persisted over the past 5 years despite the intense efforts to identify possible systematics in the data [4–9]. It is, therefore, becoming increasingly likely that it is due to new physics and an extension of ΛCDM is required to explain this mismatch [10,11]. There are three main assumptions in the foundations of this Hubble crisis problem:

- The sound horizon scale is properly calibrated by the CMB anisotropy spectrum and/or BBN in the context of standard prerecombination physics.
- The form of $E(z)$ is consistent with (1) as constrained by low $z$ distance probes which are independent of any type of calibration.
- The calibration of SnIa implemented e.g., via Cepheid stars at $z < 0.01$ remains valid at $z > 0.01$ where the Hubble flow is probed.

Corresponding to these assumptions, there are three classes of models that attempt to address the Hubble crisis by relaxing one of them. The first assumption could be violated by non-standard prerecombination physics that would decrease the sound horizon scale $r_s$ (using e.g., Early Dark Energy [10,12,13]) thus increasing the value of $H_0$ which is degenerate with $r_s$ through the measured product $H_0 r_s$. The second assumption could be violated by considering evolving dark energy models that deform $E(z)$ in comparison with $\Lambda$CDM [10,14]. Models that violate any of these two assumptions can address the Hubble tension by increasing the $H_0$ value measured by the sound horizon standard ruler, but they tend to worsen the fit to other cosmological data (especially the growth tension data) in comparison with the standard model $\Lambda$CDM [15–17].

### 1.1. Gravitational Transition as a Proposed Solution to the Hubble Crisis

The third assumption could be violated by assuming a transition of the intrinsic luminosity of SnIa to a higher value (lower absolute magnitude $M$) for $z > z_t$ (with $z_t \lesssim 0.01$)[1] compared to the luminosity at very recent cosmological times $z < z_t$ [19–21].

The transition in the intrinsic luminosity of SnIa can be connected with a gravitational transition via the expression

$$L \sim G_{\text{eff}}^{\alpha} \tag{2}$$

where $\alpha$ is a parameter determined by the detailed mechanism of the SnIa explosion. The simplest modified gravity theory that is consistent with a time variation of G is a minimal scalar-tensor theory with a step-like scalar field potential $V(\phi)$ and/or with an abrupt feature in the functional form of the nominal coupling function $F(\phi)$ [9,22,23]. In this theory, the strength of gravitational interaction is determined by $G_{\text{eff}} = \frac{1}{8\pi F} \frac{2F + 4F_{,\phi}^2}{2F + 3F_{,\phi}^2}$ while the background expansion is controlled by the Planck mass which corresponds to $G_* = \frac{1}{8\pi F}$. In practice, however, $G_{\text{eff}}$ and $G_*$ are expected to be very similar since the first derivative of the non-minimal coupling is constrained to be small by solar system constraints at zero redshift $z = 0$ and by other astrophysical probes at specific low $z$ ranges. Thus, even though different probes constrain either $G_{\text{eff}}$ or $G_*$, the constraints on one would imply similar constraints for the other [24,25]. The strength of gravity in SnIa relevant for their absolute luminosity is clearly $G_{\text{eff}}$.

Variations in $G_{\text{eff}}$ and/or $G_*$ can be constrained using a range of observations (see Table 1). Such constraints may be imposed using, e.g., Paleontology measurements [26] obtained using the age of bacteria and algae, the Hubble diagram of SnIa [27] measurement taken by luminous red galaxies or a measurement obtained using up to date primordial element abundances, cosmic microwave background as well as nuclear and weak reaction rates [28]. Demanding a transition of the SnIa absolute magnitude $M$ by $\Delta M \simeq 0.2$ after the transition (for $z < z_t$) for the resolution of the Hubble tension [19] while respecting the constraints of Table 1, the allowed range of $\alpha$ may be found $|\alpha| \in [\alpha_{min}, +\infty)$ with $\alpha_{min} \simeq 1.4$ [29] (the $+\infty$ corresponds to the $\Lambda$CDM /GR case where $G_{\text{eff}}$ is allowed (almost) zero change while $M$ changes for the required transition by $\Delta M \simeq 0.2$ [29]).

Using Equation (2) and demanding an absolute magnitude change $\Delta M = 0.2$ for $z < z_t$ (after the transition), we may find the general relation that connects the required fractional change of $G_{\text{eff}}$ after the transition defined as $\mu \equiv \frac{G_{\text{eff}-\text{aft}}}{G_{\text{eff}-\text{bef}}}$.

**Table 1.** Solar system, astrophysical and cosmological constraints on the evolution of the gravitational constant. Methods with star (∗) constrain $G_*$ (connected with the Planck mass) while the rest constrain $G_{eff}$. The latest and strongest constraints are shown for each method (updated from Ref. [30]).

| Method | $\left.\frac{\Delta G_{eff}}{G_{eff}}\right\|_{max}$ | $\left.\frac{\dot{G}_{eff}}{G_{eff}}\right\|_{max}$ (yr$^{-1}$) | Time Scale (yr) | References |
|---|---|---|---|---|
| Lunar ranging | | $1.47 \times 10^{-13}$ | 24 | [31] |
| Solar system | | $4.6 \times 10^{-14}$ | 50 | [32] |
| Pulsar timing | | $3.1 \times 10^{-12}$ | 1.5 | [33] |
| Orbits of binary pulsar | | $1.0 \times 10^{-12}$ | 22 | [34] |
| Ephemeris of Mercury | | $4 \times 10^{-14}$ | 7 | [35] |
| Exoplanetary motion | | $10^{-6}$ | 4 | [36] |
| Hubble diagram SnIa | 0.1 | $1 \times 10^{-11}$ | $\sim 10^8$ | [27] |
| Pulsating white-dwarfs | | $1.8 \times 10^{-10}$ | 0 | [37] |
| Viking lander ranging | | $4 \times 10^{-12}$ | 6 | [38] |
| Helioseismology | | $1.6 \times 10^{-12}$ | $4 \times 10^9$ | [39] |
| Asteroseismology | | $1.2 \times 10^{-12}$ | $1.1 \times 10^{10}$ | [40] |
| Gravitational waves | 8 | $5 \times 10^{-8}$ | $1.3 \times 10^8$ | [41] |
| Paleontology | 0.1 | $2 \times 10^{-11}$ | $4 \times 10^9$ | [26] |
| Globular clusters | | $35 \times 10^{-12}$ | $\sim 10^{10}$ | [42] |
| Binary pulsar masses | | $4.8 \times 10^{-12}$ | $\sim 10^{10}$ | [43] |
| Gravitochemical heating | | $4 \times 10^{-12}$ | $\sim 10^8$ | [44] |
| Strong lensing | | $3 \times 10^{-1}$ | $\sim 10^{10}$ | [45] |
| Big Bang Nucleosynthesis ∗ | 0.05 | $4.5 \times 10^{-12}$ | $1.4 \times 10^{10}$ | [28] |
| Anisotropies in CMB ∗ | 0.095 | $10^{-13}$ | $1.4 \times 10^{10}$ | [46–48] |

We thus find

$$\mu \equiv \frac{G_{eff-aft}}{G_{eff-bef}} = 10^{-\frac{2\Delta M}{5\alpha}} \tag{3}$$

For $\Delta M = 0.2$ and a SnIa absolute luminosity $L$ proportional to the Chandrasekhar mass $M_C$ (this is the simplest but not necessarily the correct assumption) we have $\alpha = -3/2$ leading to a 10% required change of $G_{eff}$ for the resolution of the Hubble tension [19]. Therefore, such a transition could be achieved by a sudden change in the value of Newton's constant by about 10% at $z_t \lesssim 0.01$ from a lower value at early times to a higher value at very recent times (during the last 50–150 Myrs) [19,20]. The actual value of $\alpha$ depends on the detailed physics of SnIa [49]. For example, if the detailed physics of SnIa were such that $\alpha = -5$, Equation (3) implies that a 4% change in $G_{eff}$ would be sufficient to resolve the Hubble crisis leading to $\Delta M = 0.2$.

On a spatial scale, based on the Hubble law, it is anticipated that such an event would change the physical properties of astrophysical objects at distances larger than a critical distance $D_t \in [15 \text{ Mpc}, 40 \text{ Mpc}]$ [30,50]. On a temporal scale, this corresponds to a transition time $t_t$ in the range $t_0 - t_t \in [50 \text{ Myrs}, 150 \text{ Myrs}]$ where $t_0$ is the present time. Since $H(z) \sim G_{eff}^{1/2}$, such a transition would also affect the Hubble expansion rate at $z < z_t$. This redshift range, however, is outside the Hubble flow which starts at $z > 0.01$ and it is hard to detect [51].

Models that are based on this gravitational transition hypothesis have the following advantages over the models that violate the first two assumptions.

- They have the same good quality of fit as the standard $\Lambda$CDM for geometric cosmological data that probe the Hubble expansion rate $H(z)$ while being consistent with local calibrators (e.g., Cepheid stars) of the SnIa absolute magnitude [20].
- They have better quality of fit than standard $\Lambda$CDM for dynamical cosmological data that probe the growth rate of cosmological perturbations (weak lensing [52–54], redshift space distortions [55–57] and cluster count data [58–63]). These data suggest weaker growth than that implied by $\Lambda$CDM in the context of general relativity [64–67]. This weaker growth is naturally provided in the context of the gravitational transition to weaker gravity (lower $G_{eff}$) at early times assumed in this class of models [19].

- The sudden gravitational transition hypothesis has several profound observational consequences that make it testable by a wide range of data on scales starting from geological and solar system scales up to astrophysical and cosmological scales. Surprisingly, current data can not rule out such gravitational transition because only the time derivative of $G_{\text{eff}}$ is strongly constrained while constraints on a transition are much weaker (see Table 1). Instead, hints have recently been found for such a transition in Tully–Fisher data [30] and also in Cepheid-SnIa calibrator data [50].

Physical mechanisms that could induce an ultra-late gravitational transition include a first order scalar tensor theory phase transition from an early false vacuum corresponding to the measured value of the cosmological constant to a new vacuum with lower or zero vacuum energy [50]. Such a transition would have many common features, such as the new early dark energy [13] first order transition proposed to take place just before recombination decreasing the scale of the sound horizon in order to increase the CMB predicted value of $H_0$. An alternative mechanism leading to a gravitational transition could involve a pressure non-crushing cosmological singularity in the recent past [68].

Most current constrains of possible evolution of $G_{\text{eff}}$ limit the time derivative of $G_{\text{eff}}$ using data from particular scales and times. The change in $G_{\text{eff}}$ since the cosmological times corresponding to the data until today, is usually inferred by assuming a smooth time variation of $G_{\text{eff}}$ since those times. Thus, these constraints are not applicable if a sudden transition of $G_{\text{eff}}$ takes place between the time of the data and the present time. Even in this context, however, a variation in $G_{\text{eff}}$ by about 10% between late and early times appears to be consistent with the current constrains shown in Table 1.

*1.2. Observational Constraints on the Gravitational Transition*

As shown in Table 1, the time variation in $G_{\text{eff}}$ can be probed by a wide range of data including solar system tests, pulsar timing, SnIa, heliosismology, paleontology, gravitational waves, CMB and BBN. The strongest constraints on a possible change in $G_{\text{eff}}$ between early and late times comes from the BBN combined with the CMB which constrain the change in $G_{\text{eff}}$ to a level $\Delta G/G \lesssim 0.05$ at $\sim 2\sigma$ level [28]. This constraint, however, is based on the Hubble expansion rate at early times which is indirectly connected with $G_{\text{eff}}$ [22] and does not directly constrain the strength of the gravitational interactions. In most modified gravity theories, the value of $G_N$ that affects the cosmological expansion rate ($H(z) \sim G_N^{1/2}$) is connected with the Planck mass as is not always identical with the parameter that determines the strength of the gravitational interactions $G_{\text{eff}}$ [22]. Thus, the CMB-BBN constraints [28,48] do not necessarily apply for $G_{\text{eff}}$ but only for $G_N$. In fact, very few studies have searched for a transition of $G_{\text{eff}}$ at very recent cosmological times.

Such an analysis was performed by [30] using Baryonic Tully–Fisher (BTF) data involving the connection between velocity rotation curves of galaxies vs their mass. According to the BTF relation, the baryonic mass of a galaxy $M_B$ is connected with its rotation velocity $v_{rot}$ as

$$M_B = A_B v_{rot}^s \tag{4}$$

where $A_B \simeq 50 M_\odot$ km$^{-4}$s$^4$ [69], $s \simeq 4$ and $A_B$ is a parameter that depends on $G_{\text{eff}}$ as $A_B \sim G_{\text{eff}}^{-2}$. In [30], the authors recently demonstrated using an up to date Tully-Fisher dataset that the best fit value of the parameter $A_B$ obtained for galaxies with distance less than about 20 Mpc differs from the corresponding value obtained for galaxies with distance larger than 20 Mpc at the $3\sigma$ level. This difference is consistent with a transition of $G_{\text{eff}}$ by 10% about 80 Myrs ago which would be consistent with the one required for the resolution of the Hubble crisis.

A recent analysis [50] has also indicated that if such a transition is allowed in the analysis of the Cepheid calibrator data, then not only is it favored by the data, but it also leads to a resolution of the Hubble crisis by decreasing the best fit SnIa absolute magnitude $M$ to a value consistent with the inverse distance ladder approach where $M$ is calibrated by the CMB implied sound horizon scale.

### 1.3. Effects of a Gravitational Transition on the Solar System Chronology

Such a gravitational transition could have also left an imprint on the temperature evolution of Earth. Teller was the first to show that the temperature of Earth is connected with the gravitational constant as follows [70],

$$T_{Earth} \sim G^{2.25} M_{\odot}^{1.75} \tag{5}$$

This was shown by computing the radius of the Earth orbit in Newtonian mechanics, taking into account that the Earth mean temperature is proportional to the fourth root of the energy received by the Sun, and assuming the conservation of angular momentum. Therefore, if we consider $M_{\odot}$ to be constant, a sudden increase in the gravitational constant a few tens of Myrs ago could mark an increase in Earth's temperature. Should this gravitational transition have happened approximately 50–70 Myrs ago, as the Tully–Fisher and Cepheid data seem to indicate, it would coincide with the era of the Paleocene–Eocene Thermal Maximum (PETM), shown in Figure 1. We stress, however, that there is a large number of factors that can affect the temperature variations shown in Figure 1, including volcanic activity and several other effects with long term consequences. In this context, this Figure cannot be considered as showing a hint for a transition of G, but only as a consistency test. Figure 1 indicates that there have been large temperature variations during the last 70 Myrs on the planet with a clear global peak about 65–50 Myrs ago. In the context of the significant changes that occurred on Earth during the last 100 Myrs, a gravitational transition would be consistent but not necessarily responsible for such changes.

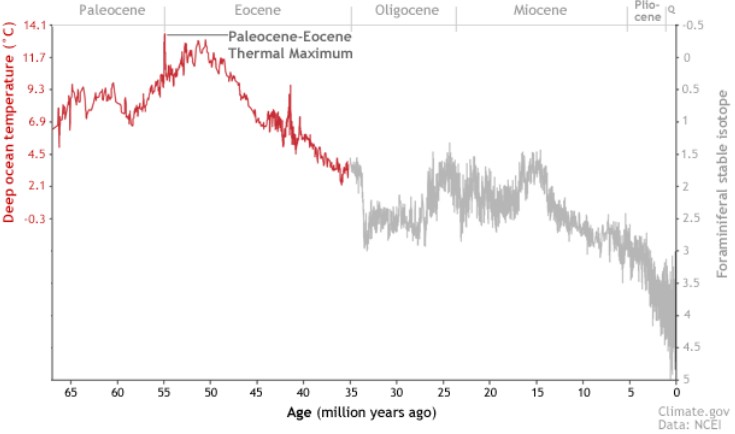

**Figure 1.** The deep ocean temperatures versus time, extended over a range of epochs from the Paleocene to the Pliocene. The temperatures in the distant past were calculated using oxygen isotope ratios from fossil foraminifera (one-celled protists, eukaryotic organisms), while the red part of the plot assumes an ice-free ocean and the gray part does not. A thermal maximum approximately 55 Myrs ago is clearly seen. The graph is by Hunter Allen and Michon Scott, using data from the NOAA National Climatic Data Center, courtesy of Carrie Morrill. For details see https://www.climate.gov/news-features/features/models-and-fossils-face-over-one-hottest-periods-earths-history (accessed on 20 March 2022).

Thus, geo-chronology and solar system chronology also constitute important sources of data that could constrain a possible 10% gravitational transition about 100 Myrs ago. There are indications for peculiar events taking place at the solar system within the last 110 Myrs.

For example, terrestrial craters found in Europe, North America and Australia indicate that the collision rate of diameter $D > 1$ km projectiles has increased by up to a factor of 3 during the past 100 Myrs [71–76]. Similar indications are obtained from lunar craters, including the 109 Myr old Tycho crater. The abundance of impact-derived glass spherules with relatively young ages found in lunar soils could also be due to a recent increase in the multi-kilometer impactor flux.

A possible cause of sudden changes in the terrestrial planet impactor flux are comet showers, produced by the passage of stars through the Oort cloud [72]. The expected duration of these showers, however, is only a few Myr [77]. This duration is not long enough to explain the observed crater and impact-spherule age distributions. The hypothesis that a large fraction of the recent (last 100 Myrs) terrestrial and lunar impactors originated from Oort cloud comets as opposed to meteorite falls of main-belt asteroids, is also supported by the fact that the composition of the largest confirmed impact crater in geo-chronology (the Chicxulub impactor) is a carbonaceous chondritic composition which is much more common for comets than main belt asteroids [78–81].

In the current analysis, we address the following question: can a 10% gravitational transition induce a factor of 2–3 observed increase in the number of LPC's reaching the solar system from the Oort cloud? A by-product of this observed increase could have been the comet that was the source of the K-T extinction event.

In order to address this question, we have implemented a Monte Carlo simulation of a large number of long period comets (LPC) with random initial positions in the Oort cloud whose initial circular orbits are perturbed by random velocity perturbations of stellar or tidal galactic origin. We thus estimate the fraction of these comets that enter the loss cone, and thus the planetary region, for various magnitudes of initial velocity perturbations, with and without a 10% increase in the strength of the gravitational interactions.

The structure of this paper is the following: In the next section, we provide background information about the Oort cloud and the LPC's that inhabit it, connecting the comet impactors with the event that caused the dinosaur extinction. In Section 3, we describe the implemented Monte Carlo simulation, and we present the results of the analysis. Concluding in Section 4, we provide some final remarks about our results and discuss possible extensions of our work.

## 2. The Oort Cloud and Long Period Comets

The Oort cloud embodies the role of a natural long period comet reservoir in our solar system. It is an outer shell of predominantly icy planetesimals that surrounds the Sun, stretching at distances ranging from approximately $10^4$ AU to $10^5$ AU, containing a population of $5 \times 10^{11}$ to $10^{12}$ objects [82], having an estimated total mass between $3.3\,M_\odot$ [83] and $38\,M_\odot$ [84].

It is believed that the vast majority of Oort cloud bodies originated between the orbits of Jupiter, Uranus, Saturn and Neptune before being forcefully ejected due to gravitational interactions. These long period comets are susceptible to gravitational perturbations from random passing stars, giant molecular clouds, etc. , leading to a constant modification of their orbits. The modifications in their orbits tend to occur more often when the comets are close to the aphelion of their approximately circular orbits, because at that point they typically have relatively small velocities. This has a significant impact on their perihelion distance, as well as their orbital inclinations. The fact that these perturbations will typically modify an Oort cloud comet's orbit many tens of thousands times over its lifespan, results in random perihelion distances and orbital inclinations, leading to an approximately spherical Oort cloud.

There are significant indications that the carbonaceous chondritic impactor responsible for the K-T mass extinction event has originated from the Oort cloud as well [85]. The alternative origin for this event would be a main-belt asteroid with a diameter > 10 km striking the Earth. In the context of a main-belt asteroid origin, such an event, however, occurs approximately once per 350 Myrs [86,87]. This leads to a reduced probability for such an origin of the K-T extinction event. This probability is further reduced if one takes into account the observed composition of such events which further reduces the impact rate of >10 km asteroids from the main belt to once per 3.8 Gyrs [71]. This renders unlikely the case that the Chicxulub crater (believed to be the cause of the K-T extinction), was formed by such an event. On the contrary, the observed carbonaceous chondritic composition of the Chicxulub crater seems to be quite common for LPC's, matching the composition of

the K-T impactor, and the rate of LPC impacts on Earth could be well within the timescale observed in the context of a stimulating mechanism enhancing their impact rate during the last 100 Myrs .

As mentioned, there are several known mechanisms that could lead to a comet being detached from the Oort cloud. More specifically, a comet's orbit can be heavily perturbed by a stellar passage close or through the Oort cloud [88], by a close encounter with giant molecular clouds [89] or by the galactic tidal force [90,91]. In the first case, we have an almost stochastic temporary process. This could lead to a large amount of comets being flung in the planetary region. The same holds true for comet encounters with giant molecular clouds, due to the fact that they could lead to mass increase and erosion. However, these events are exceedingly rare and not well understood. Lastly, we have the galactic tidal force which leads to a continuous and steady flux of comets in the planetary region.

These perturbative effects could be amplified, increasing the impact rate of LPC's on the inner solar system planets, by a sudden abrupt gravitational transition of Newton's constant. We focus on this scenario and explore its implications in the next section.

### 3. Effects of a Gravitational Transition on the LPC Flux: A Monte Carlo Approach

The LPCs approximately follow elliptic orbits (eccentricity $0 < e < 1$) with the Sun at the primary focus. The general equation of an ellipse in Cartesian coordinates ($z = 0$) is

$$\sqrt{(2\alpha e_x - x)^2 + (2\alpha e_y - y)^2} + \sqrt{x^2 + y^2} = 2\alpha \tag{6}$$

where the semi-major axis $\alpha$ as well as the eccentricity $e$ vector coordinates $e_x, e_y$ are defined, respectively, as

$$\alpha = \frac{\mu r_i}{2\mu - r_i(v_x^2 + v_y^2)} \tag{7}$$

$$e_x = \frac{x_i}{r_i} - \frac{hv_y}{\mu} \tag{8}$$

$$e_y = \frac{y_i}{r_i} - \frac{hv_x}{\mu} \tag{9}$$

$$e = \sqrt{e_x^2 + e_y^2} \tag{10}$$

and given the initial position coordinates $x_i, y_i$, the initial velocity coordinates $v_x, v_y$ and the normalized gravitational parameter $\mu = GM$ we can define the conserved specific angular momentum of an orbiting body as

$$h = x_i v_y - y_i v_x \tag{11}$$

as well as the initial distance from the primary focus of the ellipse (see Figure 2)

$$r_i = \sqrt{x_i^2 + y_i^2}. \tag{12}$$

The dynamical equations obeyed by the comets of the Oort cloud are approximated as

$$\ddot{x} = -\frac{\mu}{r^3}x \tag{13}$$

$$\ddot{y} = -\frac{\mu}{r^3}y \tag{14}$$

$$\ddot{z} = -\frac{\mu}{r^3}z - 4\pi G\rho z \tag{15}$$

where the dots denote time derivatives, $r^2 = x^2 + y^2 + z^2$, $\rho \simeq 0.2 M_\odot$ pc$^{-3}$ is the mean galactic matter density [83] and $\mu \equiv GM_\odot$. We now consider units such that the distance $r$

is measured in AU and time $t$ is measured in years. In order to express the above system in these units we use the relations $GM_\odot = 4\pi^2 \frac{\mathrm{AU}^3}{\mathrm{yr}^2}$ and 1 pc $= 206,000$ AU. These lead to the system

$$\ddot{x} = -\frac{4\pi^2}{r^3} x \tag{16}$$

$$\ddot{y} = -\frac{4\pi^2}{r^3} y \tag{17}$$

$$\ddot{z} = -\frac{4\pi^2}{r^3} z - (4\pi)^3 z \times 5 \times 10^{-18} \tag{18}$$

where now $x, y, z$ are in AU and $t$ is in years. In view of the negligible contribution of the galactic density we approximate the system as spherically symmetric and focus on trajectories on the $xy$ plane setting $z = 0$. Given the initial conditions $x_i, y_i, v_x, v_y$, the semi-major axis $\alpha$ and the eccentricity $e$ of the elliptic orbit can be obtained from (10).

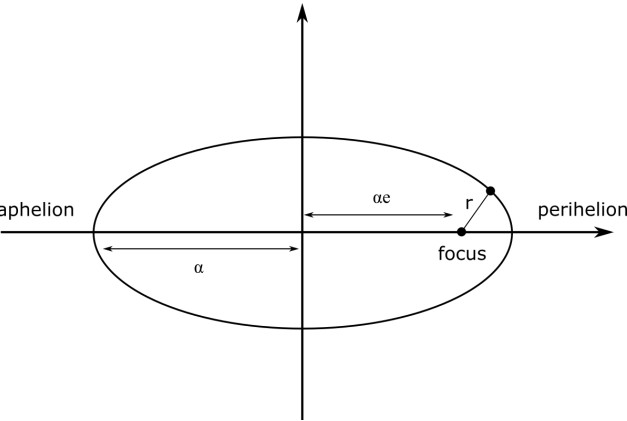

**Figure 2.** The definition of the semi-major axis ($\alpha$), eccentricity (e), the primary focus and the distance from the primary focus (r) of an ellipse.

In order to observe the impact that a gravitational transition would have on the trajectories of LPC's, we construct a simple Monte Carlo iterative process using the following steps,

- We consider a sample of $N = 10^5$ points (LPCs) with random initial radial coordinate distances $r_i$ from the primary focus of the ellipse ranging from $10^4$ AU to $4 \times 10^4$ AU, with initial velocity corresponding to circular orbits perturbed by a random velocity perturbation, and with random magnitude $v_r$ ranging from 0 to 0.14 AU/yr and direction $\theta_r$. The corresponding unperturbed circular velocities $v_c = \sqrt{4\pi^2/r_i}$ range from $v_c = 0.03$ AU/yr to $v_c = 0.06$ AU/yr. Thus, the considered velocity perturbations are of the same order as the unperturbed initial circular velocities and are assumed to be induced by stellar encounters and/or by galactic tidal effects.
- The total initial velocities of each simulated comet in polar coordinates are obtained from a superposition of the unperturbed initial circular velocity plus a random velocity perturbation,

$$v_x^i = v_r \cos \theta_r - \sqrt{\frac{4\pi^2}{r_i}} \sin \theta_i \tag{19}$$

$$v_y^i = v_r \sin \theta_r + \sqrt{\frac{4\pi^2}{r_i}} \cos \theta_i \tag{20}$$

where $v_r$ corresponds to the magnitudes of the initial random comet velocity perturbations, $r_i$ is the initial comet distances from the primary focus and the random angular position of each comet is $\theta_r \in [0, 2\pi]$. The random selection of the velocity perturba-

tion is performed using a uniform probability distribution for both the magnitude $v_r \in [v_{min}, v_{max}]$ of the velocity and its direction $\theta_r \in [0, 2\pi]$ in the plane of motion.

- From comets with the randomly perturbed velocities, we then select those that have the following properties: a. their semi-major axis as obtained from (10) is in the range $\alpha \in [10^4, 4 \times 10^4]$ AU and b. their eccentricity after the perturbation is inside the loss cone, namely they have a perihelion $p$ less than $p_* = 10$ AU (approximately Saturn's distance from the Sun). This condition corresponds to eccentricities in the range $e^2 \in [1 - 2p_*/\alpha, 1]$ [83]. This implies that these perturbed comets will enter the solar system and suffer stronger perturbations by the solar system planets, which could thus further disrupt their orbits, leading to possible impacts with planets or satellites in the solar system. The percentage of comets that enter the planetary region is thus recorded for various ranges of the velocity perturbation magnitude $v_r$.
- The above Monte Carlo simulation is repeated for a 10% increased value of Newton's constant $G_{eff}$, which corresponds to increasing the value of $\mu$ (or the value $4\pi^2$ by which $\mu$ is replaced in the AU-yr units) by the same percentage. The new fraction of comets that enter the loss cone (planetary region) is thus recorded, and its ratio is taken over the corresponding fraction obtained with the standard value of $\mu$ ($4\pi^2$). This ratio provides the excess probability that the comet will enter the loss cone after the gravitational transition to stronger gravity.

Using this Monte-Carlo approach, we have shown that the increased strength in the gravitational interaction after a gravitational transition can increase the number of perturbed comets that enter the planetary region by up to a factor of 3 for velocity perturbations that are of the same order (or somewhat larger) as the velocity of the unperturbed circular orbit. This is shown in Figure 3, where we plot the probability ratio obtained from the number of comets reaching the solar system after the gravitational transition over the corresponding number that is expected to reach the solar system before the transition. Since the random selection of the velocity perturbation is performed using a uniform probability distribution for the magnitude $v_r \in [v_{min}, v_{max}]$, we have $v_{mean} = (v_{max} + v_{min})/2$. The range of $v_r$ is shown in Figure 3 as the horizontal uncertainty for each simulation run which consisted of $10^6$ random values of $v_r$ in each range shown. For each range, the maximum and minimum fraction of comets entering the planetary region was recorded and shown as the $y$-axis-error bar of Figure 3.

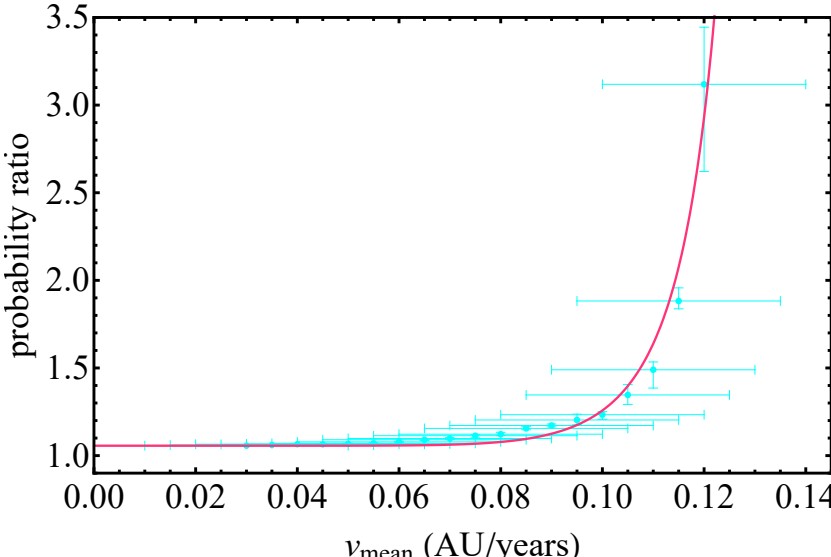

**Figure 3.** The cyan points correspond to the ratio of the probability for comets to enter our Solar system after the 10% gravitational transition, over the same probability before the transition, for different values of random initial velocity perturbations ($v_{mean}$) on the initial comet velocity. The red line corresponds to a non linear fit of the numerical results in the form of an exponential function.

The 300% increase of the probability ratio corresponds to a comet random velocity perturbation with magnitude in the range $[0.10, 0.14]$ AU/yr. By studying the effects of the gravitational transition on the probability ratio for a variety of initial velocity ranges, we find that the probability ratio is an exponentially increasing function of the mean velocity perturbation ($v_{mean}$).

In Figure 4, we show the eccentricity-semi major axis plots of the comets that display elliptic trajectories both before (left panel) and after (right panel) the gravitational transition, in the case of the initial random velocity perturbation range $v_r \in [0.1, 0.14]$. It is evident that the number of comets that reach the planetary region (green points) has nearly tripled after the gravitational transition, even though it is still very small compared to the number of comets that remain outside the planetary region.

In Figure 5, we show the initial velocity perturbation versus the initial positions of the comets that have elliptic trajectories before (left panel) and after (right panel) the gravitational transition. The blue points correspond to the comets that enter the planetary region due to the velocity perturbation. In Figure 6, we show the initial random positions of the comets in the $x$-$y$ plane colored by the value of a third parameter, their total initial velocity.

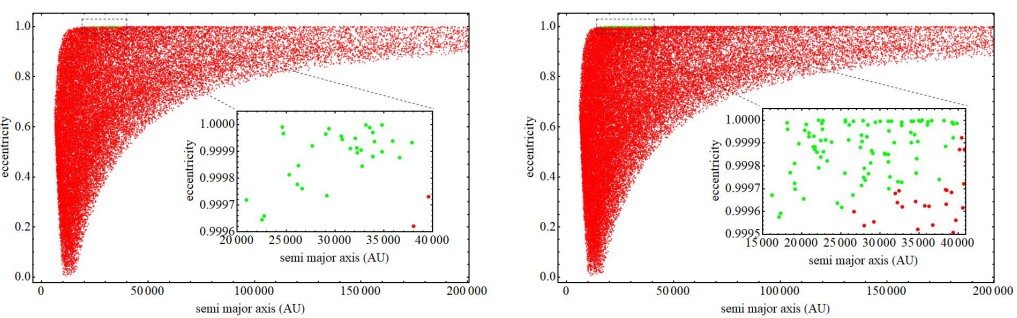

**Figure 4.** The eccentricity versus the semi major axis plots of the comets constructed in the Monte Carlo simulations for velocity perturbation magnitude in the range between 0.1 and 0.14. The red points correspond to the comets whose elliptic trajectories remain outside the planetary region after the velocity perturbation, whereas the green points correspond to those that enter the planetary region after the velocity perturbation. *Left panel:* The perturbed comet orbit parameters before the gravitational transition. *Right panel:* The perturbed comet orbit parameters after the gravitational transition has occurred, increasing the gravitational strength $\mu$ by 10%. Notice the significant increase in the number of comets that enter the solar system.

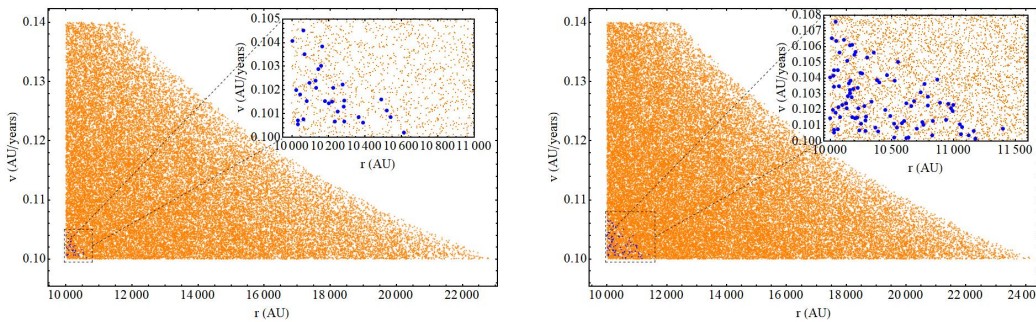

**Figure 5.** The velocity perturbation magnitude vs the initial position plots of the comets in the Monte Carlo simulations that are initially in stable orbits staying at perihelion distance larger than 10 AU. The orange points correspond to the comets whose elliptic trajectories remain outside the solar system despite the velocity perturbation, whereas the blue points correspond to those that enter our solar system after the same initial velocity perturbation. *Left panel:* The comet initial phase space coordinates before the gravitational transition. *Right panel:* The comet initial phase space coordinates after the gravitational transition has occurred, increasing the gravitational strength $\mu$ by 10%. Notice the significant increase in the number of comets that enter the solar system.

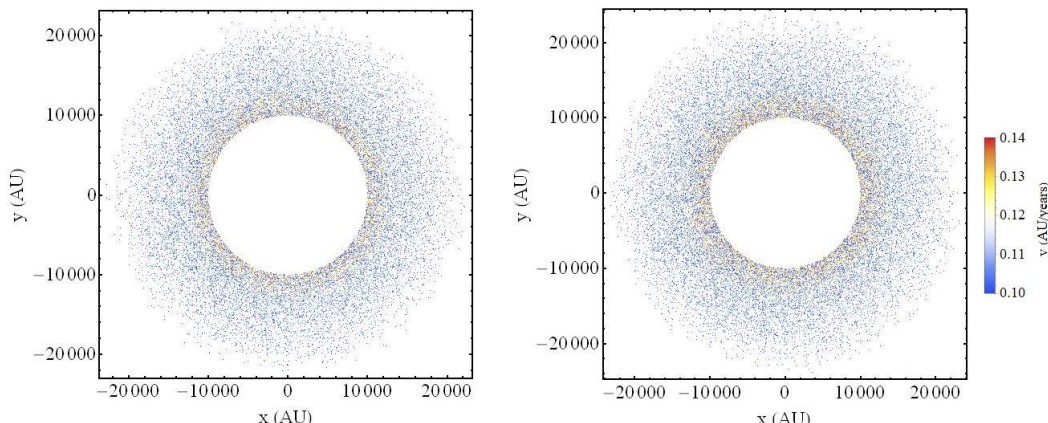

**Figure 6.** *Left panel:* The 2D scatter plot of the positions of the collection of points used in the Monte Carlo analysis (excluding those that evolve to parabolic trajectories), colored by the value of their individual velocity before the gravitational transition. *Right panel:* The 2D scatter plot of the positions of the points used in the Monte Carlo process, maintaining elliptic orbits, colored by the value of their individual velocity after the gravitational transition. The increased initial velocity is due to the increased gravitational strength, which requires higher velocity for an initial circular unperturbed orbit.

## 4. Conclusions

We have demonstrated that a sudden increase in the gravitational constant by about 10% taking place less than 100 Myrs ago can justify the observed rate of impactors on the Earth and Moon surfaces, which appears increased by a factor of 2–3 during the last 100 Myrs and may be connected with the Cretaceous–Tertiary (K-T) extinction event, eliminating 75% of life on Earth (including dinosaurs). Such a late gravitational transition event could increase by a factor of 3 or more, the number of long period comets (LPCs) entering the loss cone and impacting the planetary region from the Oort cloud due to velocity perturbations induced by stars or by the Galactic tide. In addition, it has been previously shown [19,20] that such a gravitational transition could resolve the Hubble and growth tensions of the standard cosmological ΛCDM model by increasing the SnIa absolute luminosity and weakening the growth rate of matter density fluctuations $\delta(z) \equiv \frac{\delta\rho}{\rho}(z)$ at cosmological times before the transition ($z > z_t$). Independent hints for such a transition have been pointed out in Tully–Fisher [30] and Cepheid SnIa calibrator [50] data.

If such a gravitational transition has indeed taken place, it should have left signatures in a wide range of astrophysical and geophysical data. The search for these signatures is an interesting extension of the present analysis. In particular:

- The new extended Pantheon + dataset [2] of Cepheid + SnIa data provides the opportunity for a comprehensive analysis of the unified Cepheid+SnIa data in the redshift range $z \in [0, 2.3]$ in a self-consistent and unified manner. Such an analysis which may be implemented once the full Pantheon+ dataset becomes publicly available will allow the more detailed search for signatures of a transition for redshifts $z \lesssim 0.01$, extending the analysis of [50]. Even in the current analysis of the Pantheon + dataset [2], hints for a transition are evident in Figure 10 of [2], where it is shown that the more distant Cepheids in SnIa hosts tend to have a higher value of the period luminosity parameter than the Cepheids in the closeby anchor galaxies (Figure 7). This effect is significantly amplified if the outliers are taken into account (red points).

- As implied by Equation (5), the temperature of Earth strongly depends on the value of $G_{\text{eff}}$ (see Equation (5)) [70] and so does the solar luminosity $\mathcal{L} \sim G_{\text{eff}}^7 M_{\odot}^5$. Thus, an increase in $G_{\text{eff}}$ would lead to a similar increase in the Earth temperature. Thus, a careful search of unaccounted for temperature variations of Earth during the past 150 Myrs

could either impose strong constraints on the gravitational transition hypothesis, or could reveal possible signatures of such an event.

- The search for physically motivated models and mechanisms that could generically predict the presence of such a gravitational transition realized either spatially through the nucleation of true vacuum bubbles [50] or as a transition in time involving, e.g., a pressure singularity [68], is also an interesting extension of this analysis.
- The study of the stability of the whole solar system under a gravitational transition is also an interesting extension of the present analysis. The Lyapunov time of the chaotic solar system is of $O$ (100 Myrs). Therefore, it is highly nontrivial to conclude that the solar system is stable or unstable under a finite transition of $G_{eff}$ on the timescale of the Lyapunov time. The question to address in this context is 'What is the maximum abrupt fractional change of $G_{eff}$ such that the solar system survives the transition?'. For a two body Sun–Earth system, it is easy to show via a simple simulation that the orbit of the Earth gets slightly deformed by an abrupt 10% change in the gravitational strength, but this cannot be generalized to the much more complex full solar system dynamics.

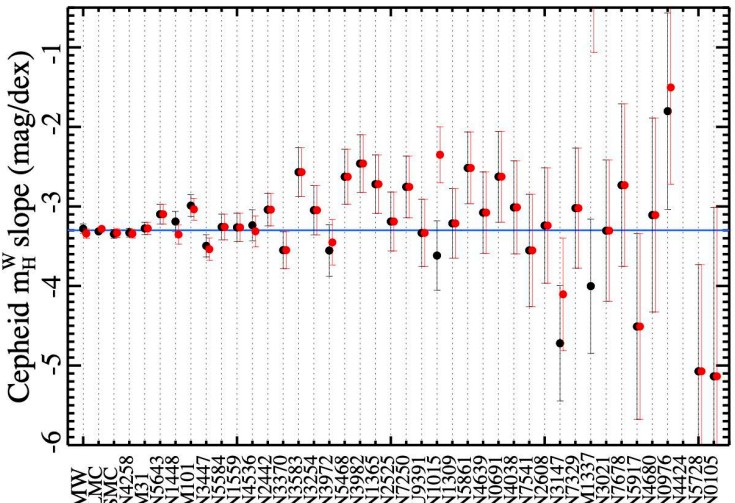

**Figure 7.** The period-luminosity parameter for the Cepheids in anchor galaxies and in SnIa host galaxies (Figure 10 from Ref. [2]). The trend for most SnIa hosts (more distant galaxies) for a higher value of the Cepheid period-luminosity parameter ($m_H^W$ slope) compared to nearby galaxies (MW, LMC, SMC, N4258 and M31) is evident. This trend is even stronger if the outliers are also included in the sample (red points).

**Funding:** This project was supported by the Hellenic Foundation for Research and Innovation (H.F.R.I.), under the "First Call for H.F.R.I. Research Projects to support Faculty members and Researchers and the procurement of high-cost research equipment Grant" (Project Number: 789).

**Data Availability Statement:** The numerical files for the reproduction of the figures can be found in the Hubble_Tension_K-T_Extinction github (https://github.com/GeorgeAlestas/Hubble_Tension_K-T_Extinction accessed on 20 March 2022.) GitHub repository under the MIT license.

**Acknowledgments:** I thank Avi Loeb, Sergei Odintsov, Vasilis Oikonomou, Ziad Zakr, Robert Helling and Ioannis Antoniou for useful comments and discussions. I also thank George Alestas for his contribution in the construction of the figures.

**Conflicts of Interest:** The author declares no conflict of interest.

## Note

1.  It has been shown by [18] that gravitational transitions with high $z_t$ are unable to resolve the cosmological tensions.

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
