# Peer review of "Is the Hubble Crisis Connected with the Extinction of Dinosaurs?"

_universe, doi:10.3390/universe8050263_

Round 1
Reviewer 1 Report
This is a very interesting paper suggesting that the hypothetical sharp change in the effective gravitational constant occuring ca. 100 Myrs ago might be responsible for the Cretacous-Tertiary extinction because of increase of the number of impactors from the Oort cloud. The paper is well written and coherent in assumptions with other astrophysical tests consistent with sharp G_eff change scenario.
I'm happy to recommend the paper to be published.
Author Response
Please see attached pdf file with the response to both reviewers.

Reviewer 2 Report
In previous works the author studied the impact of a late-time transition of the gravitational coupling on the physics at cosmological and astrophysical domains. One of the most important consequences of this transition would be the increase of the absolute magnitude of supernovae of Type Ia at z<0.01, which could alleviate or even solve the Hubble tension. The exact increase of the value of G needed to reconcile the SH0ES and CMB measurements of H0 depends on the parameter α of formula (2), which in turn depends on the physical mechanism that governs the SNIa explosions. If |α| is of order one G needs to increase by ∼10%. The author studies how this increase affects the impact rate of kilometer sized objects in the last ∼100 Myr in the vicinity of the Sun (<10 AU). This is a very original idea. The results show that actually a late-time transition of G leads to a larger flux of objects coming from the Oort cloud, something that could be linked to the Chicxulub impactor event. Before recommending this paper for publication, though, I would like to raise a couple of questions and (minor) comments:
Question 1: What is the relation between the local and cosmological values of G? Is the G employed to compute the absolute luminosity of SNIa the same that appears in the cosmological Poisson equation? How is it related to the G that enters Friedmann equation?
Question 2: The results plotted in Fig. 3 depend, of course, on the mean random velocity perturbation. How are the x- and y-error bars computed? Once the author selects a value of \nu_{mean}, how does he generate the random velocity perturbations? Does he use a Gaussian distribution centered at \nu_{mean}? With what standard deviation?
Comment 1: It is not clear to me up to what extent the Paleocene-Eocene thermal maximum shown in Fig. 1 can be considered a hint for the transition of G. There is also a similar peak in the Eocene era, which happened ∼40 Myr ago.
Comment 2: Some references in Table 1 are not updated. For instance, for the cosmogical constraints on G the author cite the paper [42], which is from 2009. Maybe the author could cite arXiv:1303.4330, or even the more recent work arXiv:2006.04273. In these papers the authors obtain their constraints in the context of Brans-Dicke theory with a constant potential.
Typo 1: In page 2, first line: calirated --> calibrated
Typo2: In page 10 it is written "The corresponding unperturbed circular velocities sqrt{4\pi / r}". I think it should be sqrt{4\pi^2 / r}.
I will be glad of carrying out the review of the second version of the paper when it is available.
Author Response

(The authors gave the same response as above.)

Round 2
Reviewer 2 Report
The author has duly addressed all the suggestions raised in my first report. I only have one more comment. In the new text of page 2, he says: "Thus, even though different probes constrain either Geff or G∗ the constraints on the one would imply similar constraints for the other". It would be useful for the reader if he could add some references to support his statement. For instance:
- Nature 425 (2003) 374-376
- JCAP 09 (2021) 040 [arXiv:2105.14819]
Once the author addresses this point I will be glad to accept his very interesting paper for publication.
Author Response
I thank the referee for pointing out these useful references which have been included in the new version of the paper at the point suggested by the referee.